# Oncologic Benefits of Neoadjuvant Treatment versus Upfront Surgery in Borderline Resectable Pancreatic Cancer: A Systematic Review and Meta-Analysis

**DOI:** 10.3390/cancers14184360

**Published:** 2022-09-07

**Authors:** Hye-Sol Jung, Hyeong Seok Kim, Jae Seung Kang, Yoon Hyung Kang, Hee Ju Sohn, Yoonhyeong Byun, Youngmin Han, Won-Gun Yun, Young Jae Cho, Mirang Lee, Wooil Kwon, Jin-Young Jang

**Affiliations:** 1Department of Surgery and Cancer Research Institute, College of Medicine, Seoul National University, Seoul 03080, Korea; 2Department of Surgery, Korea University Medical Center, Seoul 136-701, Korea; 3Department of Surgery, Chung-Ang University Gwang-Myeong Hospital, Gwangmyeong 14353, Korea; 4Department of Surgery, Uijeongbu Eulji Medical Center, School of Medicine, Eulji University, Daejeon 34824, Korea

**Keywords:** meta-analysis, pancreatic cancer, borderline resectable, neoadjuvant, prognosis

## Abstract

**Simple Summary:**

Borderline resectable pancreatic cancer (BRPC) has been primarily indicated for neoadjuvant treatment (NAT) in the last decade. This study is the updated meta-analysis for only patients with BRPC including recent NAT regimens such as FOLFIRINOX. The OS, R0 resection rate, and node-negativity rate was improved in NAT group compared with upfront surgery. Providing high-quality evidence is important to standardize the treatment protocol and help physicians decide the appropriate pancreatic cancer treatment.

**Abstract:**

Neoadjuvant treatment (NAT) followed by surgery is the primary treatment for borderline resectable pancreatic cancer (BRPC). However, there is limited high-level evidence supporting the efficacy of NAT in BRPC. PubMed was searched to identify studies that compared the survival between BRPC patients who underwent NAT and those who underwent upfront surgery (UFS). The overall survival (OS) was compared using intention-to-treat (ITT) analysis. A total of 1204 publications were identified, and 19 publications with 21 data sets (2906 patients; NAT, 1516; UFS, 1390) were analyzed. Two randomized controlled trials and two prospective studies were included. Thirteen studies performed an ITT analysis, while six presented the data of resected patients. The NAT group had significantly better OS than the UFS group in the ITT analyses (HR: 0.63, 95% CI = 0.53–0.76) and resected patients (HR: 0.68, 95% CI = 0.60–0.78). Neoadjuvant chemotherapy with gemcitabine or S-1 and FOLFIRINOX improved the survival outcomes. Among the resected patients, the R0 resection and node-negativity rates were significantly higher in the NAT group. NAT improved the OS, R0 resection rate, and node-negativity rate compared with UFS. Standardizing treatment regimens based on high-quality evidence is fundamental for developing an optimal protocol.

## 1. Introduction

Pancreatic cancer is the seventh-leading cause of cancer-related death worldwide, and has a poor prognosis and low resection rate [1,2]. Surgical resection with adjuvant chemotherapy is the standard treatment for pancreatic cancer [3]. However, even if a patient undergoes surgery with curative intent, their survival is not dramatically improved. Although microscopically margin-negative resection (R0 resection) is achieved, early recurrence develops through micrometastases that occurred prior to surgery without being detected [4,5,6,7]. These clinical experiences and theoretical backgrounds imply that PC is a systemic disease [8].

A new therapeutic approach known as neoadjuvant treatment (NAT), which consists of chemotherapy alone or chemotherapy with radiotherapy, was proposed to improvepatients’ survival by obtaining systemic control of the disease and R0 resection, which is recognized as one of the strongest prognostic factors for early disease recurrence. Borderline resectable pancreatic cancer (BRPC) has been primarily indicated for NAT to achieve an improved R0 resection rate, as it has a high potential for R1 resection despite being technically resectable.

The National Comprehensive Cancer Network guidelines recommend NAT for BRPC, although high-quality evidence is still lacking. Most studies had retrospective designs [9,10]; a few studies only evaluated patients with BRPC [10,11], or in some studies the sample size was too small in one randomized controlled trial (RCT) involving only BRPC patients [12]. Moreover, a few meta-analyses compared the treatment outcomes of BRPC patients who underwent NAT with those of patients who underwent upfront surgery (UFS).

Most meta-analyses examining the efficacy of NAT for BRPC included patients with resectable pancreatic cancer, but did not distinguish BRPC from resectable pancreatic cancer (RPC), even though the definitions, current standard care, and purposes of NAT differed from each other [13,14]. A pooled analysis that included 24 studies with 313 patients provided the patient-level outcome of NAT for BRPC patients. The patient-level median overall survival was 22.2 months and the resection rate was 67.8% in this study. However, the anti-cancer agents used for neoadjuvant chemotherapy in this study included folinic acid (leucovorin), fluorouracil (5-FU), irinotecan, and oxaliplatin (FOLFIRINOX) only [15]. Furthermore, previous investigations did not provide a comprehensive analysis according to the NAT protocol or the chemotherapy regimen.

In the last decade, NAT has become an accepted approach, particularly for patients with BRPC. Providing high-quality evidence is important to standardize the treatment protocol and help physicians decide the appropriate pancreatic cancer treatment. Therefore, this study aimed to compare the oncologic benefits of neoadjuvant chemotherapy, with or without radiotherapy, with upfront resection in patients with BRPC only.

## 2. Materials and Methods

### 2.1. Literature Search

A systematic literature search was performed in Medline (PubMed), in order to identify published articles reporting the oncological outcomes of BRPC patients treated with NAT or UFS up to 11 February 2022. A combination of heading terms, including “borderline”, “resectable”, and “pancreatic neoplasms”, were used during the database search. The relevant keywords were “neoadjuvant”, “surgery”, and “pancreatic cancer”. This systematic review was performed in accordance with the Preferred Reporting Items for Systematic Reviews and Meta-Analyses Standard Guidelines. This systematic review was registered with the Research Registry (identifying number: reviewregistry1433).

### 2.2. Selection Criteria

Only publications written in English were eligible for inclusion. Case reports, case series with a sample size of less than five, review articles, editorials, and consensus proceedings were considered ineligible. The following were included in the analysis: (1) studies reporting patients with BRPC according the standard guidelines; (2) RCTs or observational studies that compared the clinical outcomes of BRPC patients who underwent NAT and those of patients who underwent UFS; and (3) studies that provided sufficient data to reveal the survival outcomes. By contrast, (1) studies that did not show the survival outcomes of patients with BRPC alone, (2) studies that did not clarify the resectability status, (3) studies that did not specify the NAT regimens used, (4) studies that lacked data for analysis, and (5) studies with overlapping data, were not included. Moreover, when a study included RPC or locally advanced pancreatic cancer (LAPC) as well as BRPC, it was excluded if it did not provide a comparison of the survival outcomes of BRPC patients alone.

### 2.3. Definition and Study Endpoint

Only patients with “borderline resectable PDAC” were included in this study. Results of the assessment of the resectability status of PC patients in each study are summarized in Table 1. The primary endpoint of this study was OS. The hazard ratios (HR) with the corresponding 95% confidence intervals (CI) for OS from the multivariate Cox proportional hazards regression models were obtained by directly describing the included publications. If the study did not provide the HRs and 95% Cis, the HRs that were calculated from the Kaplan–Meier curves according to Tierney et al. [16] were used. The secondary outcomes were resection rate, R0 resection rate, and lymph node positivity rate. The survival outcomes in the intention-to-treat (ITT) analysis and the resected group were compared. The OS was compared in subgroups according to the NAT protocol (chemotherapy or chemoradiotherapy) and neoadjuvant chemotherapy regimen (gemcitabine/S-1 based or FOLFIRINOX) that were used. 

### 2.4. Quality Assessment and Data Extraction

Relevant studies were screened and assessed according to the inclusion and exclusion criteria by two independent reviewers (H.S.J. and H.K.). RCTs and retrospective studies were included. The following data were collected: title, first author, and publication year. Any disagreement regarding the data extraction was resolved by discussion with another investigator (J.Y.J.). The choice of the articles included in this review were in accordance with the Preferred Reporting Items for Systematic Reviews and Meta-Analyses statement (PRISMA). Quality assessment of the included studies was performed using the Cochrane collaboration’s tool for RCTs (Appendix A) and the Risk of Bias Assessment Tool for Nonrandomized Studies (RoBANS) (Appendix A) for non-randomized controlled studies by two reviewers (H.S.J. and H.K.) [32].

### 2.5. Data Synthesis and Statistical Analysis

For the primary outcome, the HRs and 95% CIs for OS were estimated using an inverse variance model to synthesize the data. A random-effects model was used for the analysis of OS. The resection, R0, and LN metastasis rates among the resected patients from each included study were pooled using the inverse variance method, in order to obtain the odds ratios (ORs) with the corresponding 95% CIs. The heterogeneity between the studies was quantified using the I^2^ metric. The I^2^ values of 25%, 50%, and 75% correspond to low, moderate, and high degrees of heterogeneity, respectively. A random-effects model was used to compare the survival and resection rate, since the inherent heterogeneity of participants was due to the various definitions of the term “borderline resectable” and its treatment protocol. Meanwhile, a fixed-effects model was used to analyze the pathological outcomes. Publication bias was detected using funnel plots. All statistical analyses were performed using Review Manager (RevMan) (version 5.4; The Cochrane Collaboration, Nordic Cochrane Center, Copenhagen, Denmark).

## 3. Results

### 3.1. Study Selection

A total of 1204 publications were identified after searching the Medline database. A total of 1170 studies were excluded after screening the titles and abstracts, and 34 publications were assessed for eligibility. After a full-text assessment of 34 studies, 15 were excluded according to the exclusion criteria. Finally, 19 publications with 21 data sets were included in the analysis, involving a total of 2906 patients (NAT, 1516; UFS, 1390) (Figure 1) [6,9,10,12,17,18,19,20,21,22,23,24,25,26,27,28,29,30,31].

### 3.2. Study Characteristics

The baseline study characteristics are summarized in Table 1 and Table 2. Two RCTs [12,26] and two prospective study designs [6,10] were included in the analysis. Five studies included resectable pancreatic cancer patients [6,19,20,22,26], while one study [21,22] included patients with locally advanced and metastatic pancreatic cancer. When the study included patients with other types of pancreatic cancer and BRPC, only the BRPC patient pool was analyzed in this study. Thirteen studies (15 data sets) performed ITT analyses, [6,10,12,20,21,23,24,25,26,27,28,29,30,31], while six studies presented the data of resected patients [9,17,18,19,22,30]. With regard to the chemotherapy regimen in the neoadjuvant setting, 16 studies used gemcitabine [6,9,10,12,17,18,19,21,22,23,24,25,26,27,28,29] and 4 of those used gemcitabine plus nab-paclitaxel [6,24,26,27]. Five studies described the FOLFIRINOX-including regimen [22,28,29,30,31]. Ren et al. and Terlizzi et al. only investigated the FOLFIRINOX regimen [30,31]. As a part of NAT, radiotherapy was administered in 14 of 19 studies [6,9,12,17,18,19,20,22,23,25,28,29,30,31], while 7 studies used concurrent chemoradiation therapy (CCRT) [9,12,18,19,20,22,23].

### 3.3. Survival Difference between NAT and UFS Groups

The NAT group had significantly better OS than the UFS group in the ITT analyses (HR: 0.63, 95% CI = 0.53–0.76, I^2^ = 58%) and in resected patients (HR: 0.68, 95% CI = 0.60–0.78, I^2^ = 0%) (Figure 2A,B). 

The OS of NAT and UFS was also compared in subgroups according to the chemotherapy regimen used (gemcitabine- or S-1-based and FOLFIRINOX). Patients treated with both neoadjuvant gemcitabine- or S-1-based chemotherapy and neoadjuvant FOLFIRINOX showed improved OS in the ITT analysis (gemcitabine, HR: 0.66, 95% CI = 0.56–0.78, I^2^ = 41%; FOLFIRINOX, HR: 0.56, 95% CI = 0.29–1.06, I^2^ = 82%) (Figure 3A) and in resected patients (gemcitabine, HR: 0.70, 95% CI = 0.60–0.80, I^2^ = 0%; FOLFIRINOX, HR: 0.54, 95% CI = 0.31–0.96, I^2^ = 65%) (Figure 3B). 

In another subgroup analysis according to whether the NAT protocol included radiotherapy, OS did not significantly differ among the subgroups (*p* = 0.14) (Appendix A).

### 3.4. Survival Difference between NAT and UFS Groups

The resection rate (NAT, 67.9%; UFS, 81.4%), R0 rate (NAT, 81.7%; UFS, 58.7%), and LN positivity rate (NAT, 46.4%; UFS, 78.0%) by intention-to-treat are described in Table 2. The resection rate was higher in the UFS group in the ITT analysis (OR: 0.29, 95% CI = 0.23–0.36; I^2^ = 75%) (Figure 4A). The studies showed high heterogeneity. The R0 resection and lymph node positivity rates were obtained in 16 studies (originally reported; 14, additionally obtained; 2) and in 15 studies (originally reported; 13, additionally obtained; 2), respectively. The R0 resection rate among the resected patients in the NAT group was significantly improved (OR: 4.16, 95% CI = 3.35–5.17, I^2^ = 48%) (Figure 4B). The lymph node positivity rate among the resected patients in the NAT group was relatively lower (OR: 0.26, 95% CI = 0.21–0.32; I^2^ = 68%) (Figure 4C).

### 3.5. Publication Bias

Funnel plots of OS comparing NAT with UFS in all patients are shown in Appendix A. No significant asymmetry was observed in the funnel plots.

## 4. Discussion

The current meta-analysis with 2906 patients (NAT, 1516; UFS, 1390) mainly examined those with BRPC alone. The NAT group showed improved OS compared with the UFS group in the ITT analysis and in resected patients. No significant differences were observed in the OS according to the chemotherapy regimen or according to whether radiotherapy was included in the NAT protocol. The resection, R0 resection, and negative lymph node rates among the resected patients in the NAT group were higher.

In the present study, a comparison of OS was reported using ITT analysis, in order to reduce various types of biases according to the treatment effect. A single-arm meta-analysis that performed ITT analysis showed that NAT improved the median OS of BRPC patients (19.2 vs. 12.8 months) [13]. In our meta-analysis, comparative studies that described the survival outcomes of NAT and UFS groups only included BRPC patients, in order to perform a direct comparison. Pan et al., a recently published meta-analysis that only included comparative trials, reported a higher OS in the NAT group in the ITT analysis (HR: 0.48, *p* < 0.001) and in resected patients (HR: 0.66, *p* = 0.001), which is consistent with the primary outcome of this study [14].

Patients administered with neoadjuvant chemotherapy plus gemcitabine or S-1 and FOLFIRINOX both showed better survival outcomes in all and in resected patients in the present study. Subgroup differences according to the chemotherapy regimen used were not significant in this study. However, a relatively small number of trials and participants in the FOLFIRINOX group were included, which may have resulted in the lack of survival differences from the gemcitabine- or S-1-based subgroup. Since the chemotherapy regimen and NAT protocol vary at the study level, there was inherent heterogeneity in our study. FOLFIRINOX is the most effective regimen for patients with metastatic pancreatic cancer [33]. Furthermore, a favorable median OS with FOLFIRINOX has been reported in LAPC (24.0 months) and in BRPC (22.2 months) patients in the patient-level meta-analyses. [15,34] The PREOPANC trial revealed that NAT plus gemcitabine improved the OS compared with surgery in BRPC patients in the predefined subgroup analysis. The efficacy of multiagent use is expected to outweigh that of a single agent. The ongoing PREOPANC-2 trial, which directly compared two neoadjuvant regimens (FOLFIRINOX versus gemcitabine-based), would provide convincing evidence supporting the efficacy of a standard chemotherapy regimen in a neoadjuvant setting [35].

In this systematic review, neoadjuvant chemoradiotherapy did not improve patients’ survival compared with neoadjuvant chemotherapy alone. The patient-level meta-analysis regarding the effect of neoadjuvant FOLFIRINOX did not find an association between the percentage of patients who underwent neoadjuvant (chemo)radiation and the median OS [15]. Additional neoadjuvant radiotherapy had little effect on the OS in the ITT analysis and in resected patients, based on the results of the meta-regression analysis [14]. Due to the heterogeneity of the NAT protocol, which was attributed to the radiotherapy regimen, radiation dose, and chemotherapy schedules, this result should be interpretated with caution, considering the theoretical influence on R0 resection and other pathologic advantages of radiotherapy. The recent Alliance A021501 trial that compared mFOLFORINOX with or without stereotactic body radiation therapy, reported that mFOLFIRINOX with hypofractionated radiotherapy did not improve the OS of patients with BRPC [36].

The resection rate was lower in the NAT group (NAT, 67.9%; UFS, 81.4%). Previous meta-analyses also showed similar resection rates for each group and substantial heterogeneity across studies [13,14,37]. This may be explained by relatively low response rates to chemotherapy of pancreatic cancer compared with other gastrointestinal cancers, and lack of reliable criteria to select patients who are suitable to undergo surgical resection in terms of technical and biological aspects. Additionally, the low rate to complete chemotherapy cycles due to various adverse effects from anti-cancer agents might result in difficulty in disease control for potential micrometastases [38]. However, the modified chemotherapeutics used in current practice, accumulated experience of treating and operating on BRPC patients, and improved surgical techniques may contribute to increased resection rates in patients who undergo NAT in the future.

The R0 resection rate (NAT, 81.7%; UFS, 58.7%) and lymph node positivity rate (NAT, 46.4%; UFS, 78.0%) were found to be significantly improved in patients with BRPC who underwent NAT. This systematic review presented the benefit of neoadjuvant therapy for the downstaging of tumors or facilitation of R0 resection. The better negative lymph node rate among patients who underwent pancreatic resection in the NAT group may be due to the effect of NAT, causing regression of the tumor and metastatic lymph nodes.

Meanwhile, several negative surgical outcomes also exist for BRPC patients. When a patient receives upfront surgery, the R0 resection rate is lower despite the higher resection rate. Furthermore, the rate of postoperative pancreatic fistula is considered to be higher in patients with upfront resection [39,40]. The greater lymph node positivity rate and diminished tumor down-staging effect might also cause a high risk of systemic recurrence after upfront surgery. For patients who undergo NAT, the lower resection rate due to disease progression during NAT, or due to complications followed by chemoradiation and the delayed recovery after surgery, may affect the surgical outcomes. Given the remaining debate on the role of NAT for BRPC patients, well-controlled RCTs with ITT analyses are needed to investigate surgical outcomes with minimal biases in the era of NAT for BRPC.

This systematic review has several limitations. Firstly, most of the publications included in this study were retrospective in nature, which caused a high risk of bias. Only two RCTs and prospective studies were included, and the remaining articles were retrospective studies. One of the two RCTs was terminated early, as it already reported the significance of the effect of NAT. Secondly, substantial heterogeneity was found in the analysis of the resection rate, R0 rate, and pathological lymph node rate, owing to the clinically varying definitions of BRPC. Thirdly, head-to-head comparisons among different types of neoadjuvant chemotherapy regimens could not be performed due to the small numbers of included trials.

## 5. Conclusions

In conclusion, our meta-analysis, which only focused on BRPC patients, demonstrated that NAT provides survival benefits compared with UFS. Standardizing treatment regimens based on high-quality evidence is fundamental for developing an optimal protocol to improve patients’ survival. Furthermore, the efficacy of NAT should be prospectively explored in resectable or locally advanced PDAC, considering the dramatic advantages of preoperative chemoradiotherapy.

## Figures and Tables

**Figure 1 cancers-14-04360-f001:**
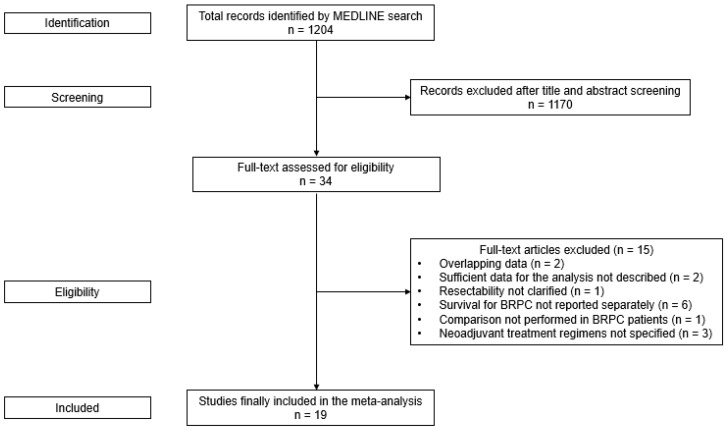
Flow diagram for selection of relevant clinical studies.

**Figure 2 cancers-14-04360-f002:**
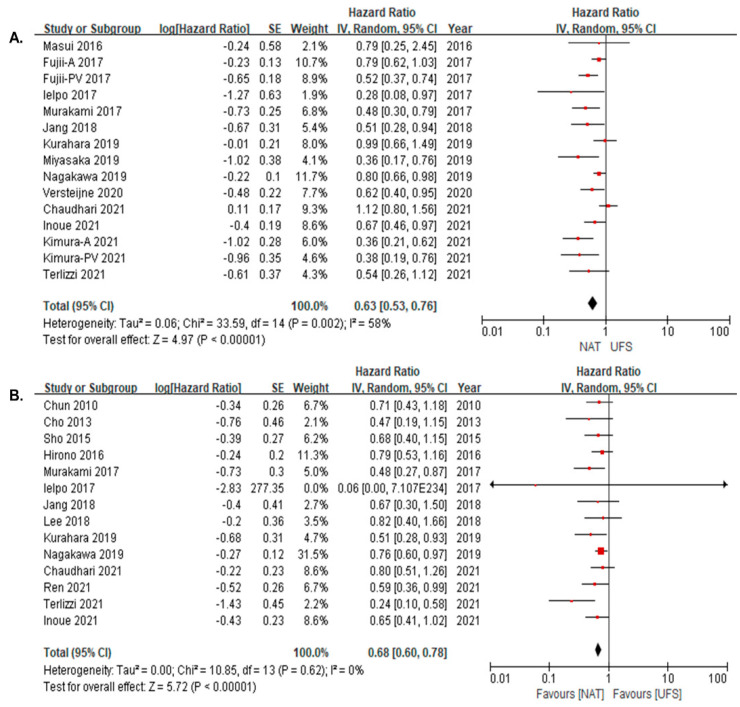
Overall survival of NAT versus UFS (**A**) in intention-to-treat analysis [6,10,12,20,21,23,24,25,26,27,28,29,31]; (**B**) in resected group [6,9,12,17,18,19,21,22,23,25,27,28,30,31]. Red squares correspond to individual studies. Squares size is proportional to the weight of the study while black diamonds shapes correspond to pooled studies.

**Figure 3 cancers-14-04360-f003:**
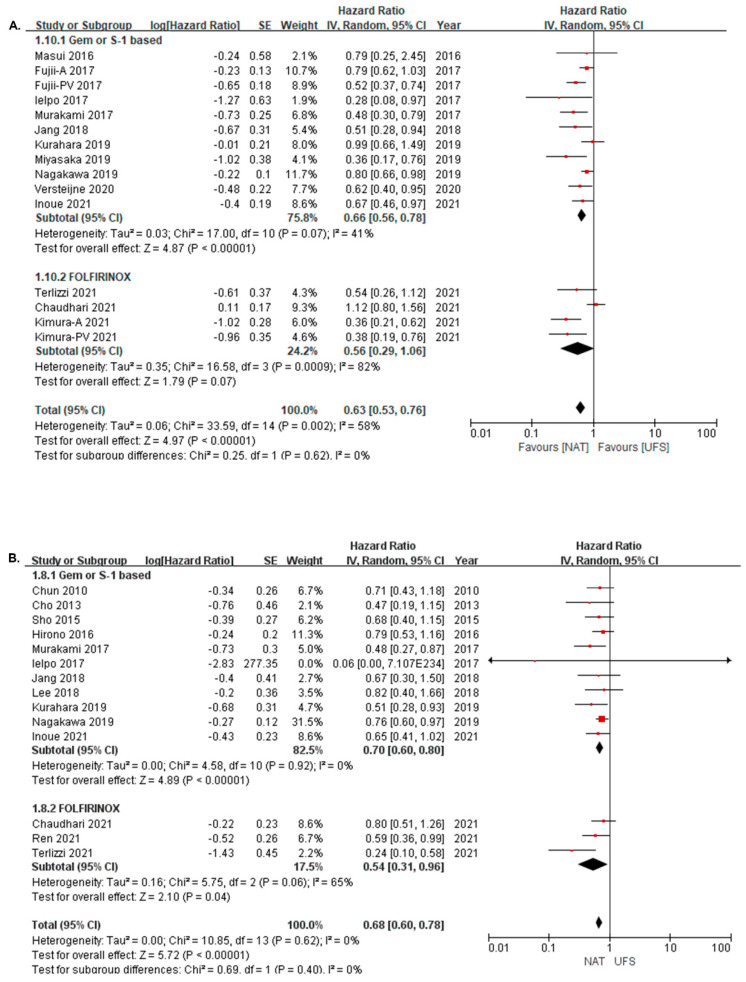
Overall survival of NAT versus UFS according to chemotherapy regimen (**A**) in intention-to-treat analysis; (1.10.1) gemcitabine- or S-1-based [6,10,12,20,21,23,24,25,26,27]; (1.10.2) FOLFIRINOX [28,29,31]; (**B**) in resected group; (1.8.1) gemcitabine- or S-1-based [6,9,12,17,18,19,21,22,23,25,27]; (1.10.2) FOLFIRINOX [28,30,31]. Red squares correspond to individual studies. Squares size is proportional to the weight of the study while black diamonds shapes correspond to pooled studies.

**Figure 4 cancers-14-04360-f004:**
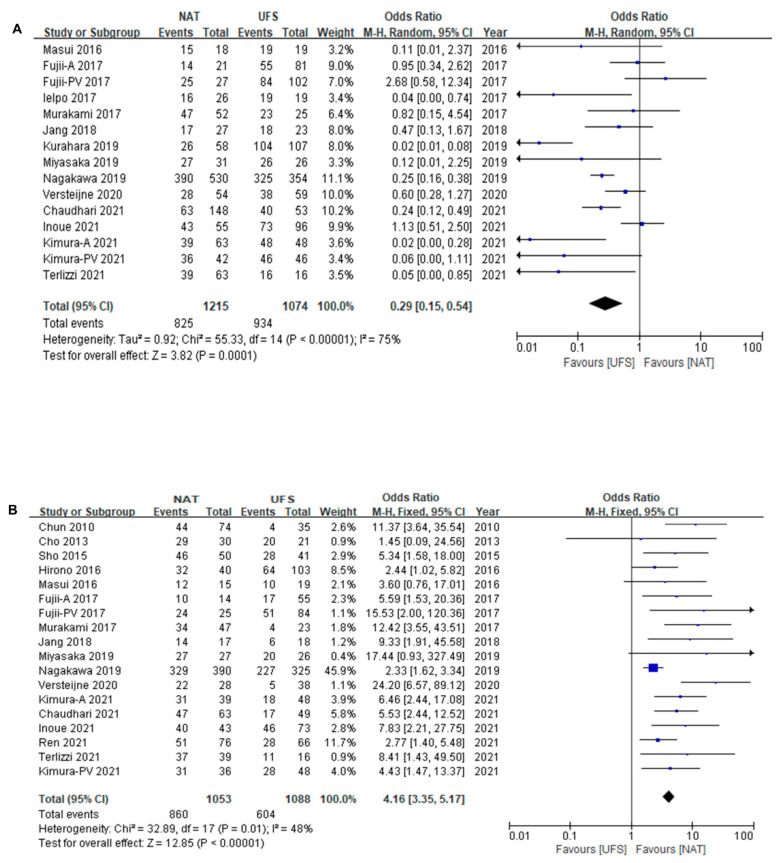
(**A**) Resection rate of NAT versus UFS in total patients. [6,10,12,20,21,23,24,25,26,27,28,29,31] (**B**) R0 resection rate of NAT versus UFS among resected patients. [9,10,17,18,19,20,21,24,25,26,27,28,29,30,31] (**C**) Pathological lymph node rate of NAT versus UFS among resected patients [9,10,12,17,18,19,20,21,24,25,28,29,30,31]. Blue squares correspond to individual studies. Squares size is proportional to the weight of the study while black diamonds shapes correspond to pooled studies.

**Table 1 cancers-14-04360-t001:** General characteristics of included studies.

Author	Country	Year	Study Period	Study Design	Resectability	Definition	Analysis	NAT Regimen
Chun [17]	US	2010	1990–2009	Retrospective	BR	Ishikawa classification	Resected	Gemcitabine-based CRT 5-FU based CRT
Cho [18]	Korea	2013	2002–2011	Retrospective	BR	MDACC	Resected	Gemcitabine-based CCRT Capecitabine in some patients
Sho [19]	Japan	2015	2006–2013	Retrospective	R+BR	NCCN	Resected	Gemcitabine-based CCRT
Hirono [9]	Japan	2016	2000–2013	Retrospective	BR	NCCN	Resected	S-1-based CCRT S-1 + Gemcitabine
Masui [10]	Japan	2016	2005–2010	Prospective	BR	Modified criteria of the study group	ITT	Gemcitabine + S-1
Fujii-Artery [20]	Japan	2017	2001–2013	Retrospective	R+BR	NCCN	ITT	S-1-based CCRT
Fujii-PV [20]							ITT	S-1-based CCRT
Ielpo [6]	Spain	2017	2007–2016	Retrospective + Prospective	R+BR	NCCN	ITT	Gemcitabine + Nab-paclitaxel IMRT since 2013
Murakami [21]	Japan	2017	2002–2015	Retrospective	BR	NCCN	ITT	Gemcitabine + S-1
Jang [12]	Korea	2018	2012–2014	RCT	BR	NCCN	ITT	Gemcitabine-based CCRT
Lee [22]	Korea	2018	2007–2014	Retrospective	R+BR+LA+M	NCCN	Resected	Gemcitabine- or 5-FU-based CCRT FOLFIRINOX
Kurahara [23]	Japan	2019	2010–2014	Retrospective	BR	NCCN	ITT	Gemcitabine- or S-1-based chemotherapy S-1-based CCRT
Miyasaka [24]	Japan	2019	2010–2017	Retrospective	BR	NCCN	ITT	Gemcitabine + Nab-paclitaxel
Nagakawa [25]	Japan	2019	2011–2013	Retrospective	BR	Japanese Classification of Pancreatic Carcinoma	ITT	Gemcitabine and/or S-1 RT (n = 319)
Versteijne [26]	Netherland	2020	2013–2017	RCT	R+BR	Dutch pancreatic cancer group criteria	ITT	Gemcitabine + Nab-paclitaxel
Inoue [27]	Japan	2021	2008–2017	Retrospective	BR	NCCN	ITT	Gemcitabine + Nab-paclitaxel
Chaudhari [28]	India	2021	2007–2019	Retrospective	BR	AHPBA/SSAO/SSO	ITT	FOLFIRINOX Gemcitabine-based RT (n = 89)
Kimura-A [29]	Japan	2021	2002–2018	Retrospective	BR	Japanese Classification of Pancreatic Carcinoma	ITT	GnP FOLFIRINOX Gemcitabine + S-1 and/or RT
Kimura-PV [29]							ITT	GnP FOLFIRINOX Gemcitabine + S-1 and/or RT
Ren [30]	US	2021	2008–2018	Retrospective	BR+LA	AHPBA/SSAO/SSO	Resected	FOLFIRINOX and/or RT
Terlizzi [31]	France	2021	2010–2017	Retrospective	BR	NCCN	ITT	FOLFIRINOX and/or RT

R, resectable; BR, borderline resectable; LA, locally advanced; NAT, neoadjuvant therapy; A, artery; PV, portal vein; ITT, intention-to-treat analysis; MDACC, MD Anderson Cancer Center; AHPBA/SSAO/SSO, American Hepato-Pancreato-Biliary Association/Society for Surgery of the Alimentary Tract/Society of Surgical Oncology; CRT, chemoradiation therapy; CCRT, concurrent chemoradiation therapy; GnP, gemcitabine plus nab-paclitaxel.

**Table 2 cancers-14-04360-t002:** Clinical outcomes of included studies.

Author	No. of Patients	ITT OS (Month)	Resected OS (Month)	Resection Rate	R0 Rate	LN Metastasis
Chun [17]	NAT 74 UFS 35	NA	23.0 15.0	74/74 (100.0%) 35/35 (100.0%)	44/74 (59.5%) 4/35 (11.4%)	20/74 (27.0%) 30/35 (85.7%)
Cho [18]	NAT 30 UFS 21	NA	45.0 23.5	30/30 (100.0%) 21/21 (100.0%)	29/30 (96.7%) 20/21 (95.2%)	9/30 (30.0%) 9/21 (42.9%)
Sho [19]	NAT 50 UFS 41	NA	24.8 16.4	50/50 (100.0%) 41/41 (100.0%)	46/50 (92.0%) 28/41 (68.3%)	15/50 (30.0%) 31/41 (75.6%)
Hirono [9]	NAT 40 UFS 103	NA	19.3 13.7	40/40 (100.0%) 103/103 (100.0%)	32/40 (80.0%) 64/103 (62.1%)	31/40 (77.5%) 77/103 (74.8%)
Masui [10]	NAT 18 UFS 19	21.7 21.1	NA	15/18 (83.3%) 19/19 (100.0%)	12/15 (80.0%) 10/19 (52.6%)	5/15 (33.3%) 5/19 (26.3%)
Fujii-Artery [20]	NAT 21 UFS 81	18.1 10.0	NA	14/21 (66.7%) 55/81 (67.9%)	10/14 (71.4%) 17/55 (30.9%)	2/14 (14.3%) 51/55 (92.7%)
Fujii-PV [20]	NAT 27 UFS 102	28.4 20.1	NA	25/27 (92.6%) 84/102 (82.4%)	24/25 (96.0%) 51/84 (60.7%)	11/25 (44.0%) 70/84 (83.3%)
Ielpo [6]	NAT 26 UFS 19	18.9 13.5	43.6 13.5	16/26 (61.5%) 19/19 (100.0%)	NA	NA
Murakami [21]	NAT 52 UFS 25	27.1 11.6	27.2 11.6	47/52 (90.45) 23/25 (92.0%)	34/47 (72.3%) 4/23 (17.4%)	34/47 (72.3%) 18/23 (78.3%)
Jang [12]	NAT 30 UFS 28	21.0 12.0	22.0 19.5	17/27 (63.0%) 18/23 (78.3%)	14/17 (82.4%) 6/18 (33.3%)	5/17 (29.4%) 15/18 (83.3%)
Lee [22]	NAT 28 UFS 45	NA	NA	28/28 (100.0%) 45/45 (100.0%)	NA	NA
Kurahara [23]	NAT 58 UFS 107	22.0 16.7	53.7 17.8	26/58 (44.8%) 104/107 (97.2%)	NA	NA
Miyasaka [24]	NAT 31 UFS 26	27.9 12.4	NA	27/31 (87.1%) 26/26 (100.0%)	27/27 (100.0%) 20/26 (76.9%)	18/27 (66.7%) 23/26 (88.5%)
Nagakawa [25]	NAT 530 UFS 354	25.7 19.0	29.8 21.5	390/530 (73.6%) 325/354 (91.8%)	329/390 (84.4%) 227/325 (69.8%)	206/390 (52.8%) 261/325 (80.3%)
Versteijne [26]	NAT 54 UFS 59	17.6 13.2	NA	28/54 (51.9%) 38/59 (64.4%)	22/28 (78.6%) 5/38 (13.2%)	NA
Inoue [27]	NAT 55 UFS 96	31.9 18.1	38.4 18.8	43/55 (78.2%) 73/96 (76.0%)	40/43 (93.0%) 46/73 (63.0%)	28/43 (65%) 57/73 (78%)
Chaudhari [28]	NAT 148 UFS 53	15 18	22 19	63/148 (42.6%) 40/53 (75.5%)	47/63 (74.6%) 17/40 (42.5%)	20/63 (31.7%) 24/40 (60%)
Kimura-A [29]	NAT 63 UFS 48	35.4 14.3	NA	39/63 (61.9%) 48/48 (100.0%)	31/39 (79.5%) 18/48 (37.5%)	17/39 (43.6%) 41/48 (85.4%)
Kimura-PV [29]	NAT 42 UFS 46	22.8 16.1	NA	36/42 (85.7%) 46/46 (100.0%)	31/36 (86.1%) 28/46 (60.9%)	17/36 (47.2%) 37/46 (80.4%)
Ren [30]	NAT 76 UFS 66	NA	35.8 27.8	76/76 (100.0%) 66/66 (100.0%)	51/76 (67.1%) 28/66 (42.4%)	30/76 (39.5%) 50/66 (75.8%)
Terlizzi [31]	NAT 63 UFS 16	29.0 27.2	63.1 27.2	39/63 (61.9%) 16/16 (100.0%)	37/39 (94.9%) 11/16 (68.8%)	8/39 (20.5%) 11/16 (68.8%)
Total	NAT 1516 UFS 1390	-	-	NAT 67.9% UFS 81.4%	NAT 81.7% UFS 58.7%	NAT 46.4% UFS 78.0%

Total proportions of resection rate, R0 rate, and LN metastasis rate were calculated as number of events divided by number of patients by intention-to-treat. ITT, intention-to-treat; OS, overall survival; NAT, neoadjuvant therapy; UFS, upfront surgery; NA, not applicable; A, artery; PV, portal vein.

## Data Availability

The data presented in this study are available on request from the corresponding author.

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
