# Peer review of "Oncologic Benefits of Neoadjuvant Treatment versus Upfront Surgery in Borderline Resectable Pancreatic Cancer: A Systematic Review and Meta-Analysis"

_cancers, 2022, doi:10.3390/cancers14184360_

Round 1

Reviewer 1 Report

"Oncologic benefits of neoadjuvant treatment versus upfront surgery in borderline resectable pancreatic cancer: a systematic review and meta-analysis" is the title of their systematic meta-analysis manuscript. The intent of Jung et al. was to demonstrate the standardization of treatment protocols based on high-quality evidence for developing the most effective protocol to increase the BRPC patient's post-operative survival.

This constitutes a large, comprehensive, and broadly coherent body of work that is appreciable.

However, there are a few minor issues stated below that can be fixed to improve the manuscript's quality.

1. Please mention any potential negative surgical outcomes for BRPC patients.

2. Please improve the figures' quality; for example, figures 2, 3, and 4 require significant improvement.

Author Response

  1. Please mention any potential negative surgical outcomes for BRPC patients.

Response> We appreciate your detailed comment to improve our manuscript’s quality. We agree that there are some negative surgical outcomes for BRPC patients. We pointed out several negatives for both treatment types (NAT and upfront surgery) as follows (Page 4, Line 287-296):

Meanwhile, several negative surgical outcomes also exist for BRPC patients. When a patient receives upfront surgery, the R0 resection rate is lower despite higher resection rate. Furthermore, the rate of postoperative pancreatic fistula is considered to be higher in patients with upfront resection (Hank et al, Uchida et al). The greater lymph node positivity rate and less tumor down-staging effect might also cause a high risk of systemic recurrence after upfront surgery. For patients who undergo NAT, the lower resection rate due to disease progression during NAT or complications followed by chemoradiation and the delayed recovery after surgery may affect the surgical outcomes. Given the remaining debate on the role of NAT for BRPC patients, the well-controlled RCTs with ITT analysis are needed to investigate surgical outcomes with minimal bias in the era of NAT for BRPC.

  1. Please improve the figures' quality; for example, figures 2, 3, and 4 require significant improvement.

Response> Thank you for your kind comment for the figures’ quality. The original version of figures summited with Power Point Files looks is uploaded with an appropriate pixel size for which this journal requested. The improved quality of figures will be provided.

Reviewer 2 Report

Major revision

1.     Why to include a study with metastatic disease ( ref 22) ??

2.     Why to include patients with other type of pancreatic cancer?  line 153-155

3.     Discussion: the authors have to discuss the resection rate ((NAT, 67.9%; UFS, 81.4%), because this is one of the main drawbacks of NAT

4.     The quality of the images in the supplementary file  requires significant improvement

Minor revision

1.     Line 51: “course” to change to “recurrence”

2.     Line 62: to define “RPC”  

3.     Line 63-65: The research results should be added

4.     Line 69: “NAT has become the mainstream treatment” -  this is correct in USA but in Europe upfront surgery still the mainstream approach. I suggest to change to “NAT has become an accepted approach”

5.     Line 96: to define “LAPC”

6.     Line 106: “Tierney et al” reference number

Author Response

Major revision

  1. Why to include a study with metastatic disease (ref 22) ??

Response> The study cited with ref 22 includes not only metastatic disease but also resectable, borderline resectable, and locally advanced diseases. Although this study is including patients pool with metastatic disease, the authors reported the separated and independent data for resection rate in BRPC patients. We requested the additional raw data for survival, R0 and node metastasis rate to the corresponding author, but we could not receive the reply. This study includes relatively large patient pool and separated BRPC from other resectability type of pancreatic cancer, so we included this study in our systematic review.   

  1. Why to include patients with other type of pancreatic cancer?  line 153-155

Response> We appreciate your detailed comment to improve our manuscript’s quality. Line 153-155 means that we did not include patients with other types of pancreatic cancer in this literature review. If an article included not only BRPC patients but also resectable or locally advanced PDAC patients, we looked into the results of the study whether the data such as survival outcome, resection rate, R0, or node metastasis rate are independently presented for only BRPC patient group. We reported the type of resectability in each study included in this systematic review in Table 1.

  1. Discussion: the authors have to discuss the resection rate ((NAT, 67.9%; UFS, 81.4%), because this is one of the main drawbacks of NAT

 Response> We are sure that your comment on the resection rate is very important to provide comprehensive information for NAT. We totally agree to discuss the lower resection rate of NAT, and we added a paragraph to discuss this point as follows (Page 4, Line 270-280):

The resection rate was lower in the NAT group (NAT, 67.9%; UFS, 81.4%). Previous meta-analyses also showed similar resection rates for each group and substantial heterogeneity across studies. [13,14,37] This may be explained by relatively low response rate to chemotherapy of pancreatic cancer compared with other gastrointestinal cancers and lack of reliable criteria to select patients who are suitable to undergo surgical resection in terms of technical and biological aspects. Additionally, the low rate to complete chemotherapy cycles due to various adverse effects from anticancer agents might result in difficulty in disease control for potential micrometastases. [38] However, the modified chemotherapeutics used in current practice, cumulated experience of treating and operating on BRPC patients and improved surgical techniques might attribute to increased resection rate for patients who undergo NAT in the future.

  1. The quality of the images in the supplementary file requires significant improvement

     Response > Thank you for your kind comment for the figures’ quality. The original version of figures summited with Power Point Files looks is uploaded with an appropriate pixel size for which this journal requested. The improved quality of figures will be provided.

Minor revision

  1. Line 51: “course” to change to “recurrence”

Response> We have changed the word “course” to “recurrence”.

  1. Line 62: to define “RPC”  

Response> “RPC” means resectable pancreatic cancer. We added the definition in Line 62-63.

  1. Line 63-65: The research results should be added

Response> We added the research results of cited study as follows (Line 64-66):

A pooled analysis including 24 studies with 313 patients provided the patient-level outcome of NAT for BRPC patients. The patient-level median overall survival was 22.2 months and the resection rate was 67.8% in this study.

  1. Line 69: “NAT has become the mainstream treatment” -  this is correct in USA but in Europe upfront surgery still the mainstream approach. I suggest to change to “NAT has become an accepted approach”

Response> We have changed the word “NAT has become the mainstream treatment” to change to “NAT has become an accepted approach”.

  1. Line 96: to define “LAPC”

Response> “LAPC” means locally advanced pancreatic cancer. We added the definition in Line 99.

  1. Line 106: “Tierney et al” reference number

Response> We added the citation “Tierney et al” in reference number 16.

Reviewer 3 Report

The meta-analysis is an important analysis of the benefits of BRPC including improvement in R0 resections, downstaging LN positivity, and improving OS. The authors appropriately point out that one of the major limitations of this type of evaluation which is the inherent bias seen with retrospective analyses. However I do not think this limitation detracts from the overall value of this work for the Pancreatic cancer community.

Author Response

Response> Thank you for your kind comment on this study. We agree with your opinion on the limitations of this systematic review. Most studies included in this meta-analysis are retrospective analyses. The authors recognized the risk of bias at the protocol step, and assessed the risk of bias for RCTs and non-RCTs with each assessment tool. We accepted the inherent bias remaining in this study and described this limitation in the discussion.

Round 2

Reviewer 2 Report

No further revision is needed